# Recent Progress and Challenges in the Diagnosis and Treatment of Gastrointestinal Stromal Tumors

**DOI:** 10.3390/cancers13133158

**Published:** 2021-06-24

**Authors:** Toshirou Nishida, Shigetaka Yoshinaga, Tsuyoshi Takahashi, Yoichi Naito

**Affiliations:** 1Department of Surgery Japan, Community Health-Care Organization Osaka Hospital, Osaka 553-0003, Japan; 2National Cancer Center, Tokyo 104-0045, Japan; 3Endoscopy Division, National Cancer Center Hospital, Tokyo 104-0045, Japan; shiyoshi@ncc.go.jp; 4Department of Gastroenterological Surgery, Osaka University Graduate School of Medicine, Osaka 565-0871, Japan; ttakahashi2@gesurg.med.osaka-u.ac.jp; 5Department of General Internal Medicine/Experimental Therapeutics/Medical Oncology, National Cancer Center Hospital East, Chiba 277-8577, Japan; ynaito@east.ncc.go.jp

**Keywords:** gastrointestinal stromal tumor, submucosal tumor, subepithelial tumor, gene panel analysis, precision medicine

## Abstract

**Simple Summary:**

Gastrointestinal stromal tumors (GIST) are potentially malignant tumors and require evidence-based surgical and/or medical treatment. Laparoscopy has similar safety and prognostic outcomes to those of laparotomy and is currently a standard procedure for localized GISTs. However, surgery for gastric GISTs less than 2 cm may be re-evaluated due to the indolent nature of the GIST and other competing risks among GIST patients. A work-up with endoscopy and endoscopic ultrasonography as well as endoscopic or percutaneous biopsy is important for the preoperative diagnosis of GISTs. Medical treatment with tyrosine kinase inhibitors is the mainstay for recurrent/metastatic GISTs. The activity of an individual drug is well correlated with gene alterations, and, in the era of precision medicine, cancer genome profiling should be considered before medical treatment.

**Abstract:**

Gastrointestinal stromal tumors (GISTs) are the most frequent malignant mesenchymal tumors in the gastrointestinal tract. The clinical incidence of GISTs is estimated 10/million/year; however, the true incidence is complicated by frequent findings of tiny GISTs, of which the natural history is unknown. The initial work-up with endoscopy and endoscopic ultrasonography plays important roles in the differential diagnosis of GISTs. Surgery is the only modality for the permanent cure of localized GISTs. In terms of safety and prognostic outcomes, laparoscopy is similar to laparotomy for GIST treatment, including tumors larger than 5 cm. GIST progression is driven by mutations in *KIT* or *PDGFRA* or by other rare gene alterations, all of which are mutually exclusive. Tyrosine kinase inhibitors (TKIs) are the standard therapy for metastatic/recurrent GISTs. Molecular alterations are the most reliable biomarkers for TKIs and for other drugs, such as NTRK inhibitors. The pathological and genetic diagnosis prior to treatment has been challenging; however, a newly developed endoscopic device may be useful for diagnosis. In the era of precision medicine, cancer genome profiling by targeted gene panel analysis may enable potential targeted therapy even for GISTs without *KIT* or *PDGFRA* mutations.

## 1. Introduction

The gastrointestinal stromal tumor (GIST) is a potentially malignant mesenchymal tumor (sarcoma) that usually expresses KIT or DOG1 proteins by immunohistochemistry (IHC). GISTs are considered to be a lineage of immature mesenchymal cells capable of differentiating into the interstitial cells of Cajal (ICC), which serve as pacemaker cells of the gastrointestinal (GI) tract [1,2,3,4]. Hence, GISTs are exclusively found in various parts of the GI tract, including the stomach (approximately 60–65%, with most found in the upper stomach), the small intestine (20–25%, mainly in the proximal small intestine, the duodenum, and proximal jejunum), and the colon, as well as in the rectum (comprising a low % of GISTs, mostly in the distal rectum), the esophagus (1%), and, rarely, in parts of the extra-GI tract, such as the peritoneum and major omentum. The majority of GISTs have a gain-of-function mutation in either *KIT* (70%) or *PDGFRA* (10–15%), and some (nearly 15%) may have other mutations in *BRAF*, *RAS* family genes, and *NF1*, and alterations in the *SDH* (succinate dehydrogenase; complex III in the mitochondrial electron transport system) complex or in *NRTK* translocation (Table 1) [4,5,6,7,8,9,10,11]. These mutations and alterations are mutually exclusive in primary GISTs.

The incidence of clinical GISTs, symptomatic GISTs or GISTs requiring treatment, is assumed to be 6–22 cases per million per year [1,2,3,4]. However, the true incidence of GISTs is more complicated and unknown because of the presence of mini-GISTs (asymptomatic GISTs less than 2 cm incidentally found by endoscopy) and of micro-GISTs (GISTs that are usually less than 1 cm and incidentally found by pathological examinations of resected specimens) [2,4,12]. Pathological examination of the stomachs and rectums of middle-aged adults reveals micro-GISTs in 10–35% and in 0.1–0.2% of cases, respectively [12,13,14,15]. Small submucosal tumors (SMTs) that are less than 2 cm are relatively frequent endoscopic findings in the stomach. It has been reported that endoscopy may reveal small neoplastic SMTs in 0.15% of middle-aged adults and that half of them are considered to be GISTs [12,16,17]. The natural history of mini-GISTs and micro-GISTs is unknown, and its clinical relevance needs to be elucidated.

## 2. Diagnosis

There are no symptoms or signs specific to GISTs. The most frequent symptoms include gastrointestinal bleeding and subsequent anemia, followed by abdominal pain, weight loss, and a palpable abdominal mass [4,9]. GISTs are unusually associated with bowel obstruction or perforation, except in cases of large tumors. It should be noted that a significant number of GISTs are asymptomatically found as SMTs by cancer-screening endoscopy or may be incidentally found in explorations of other diseases. GISTs are diagnosed from childhood to late adulthood, and the reported median age is in the 60 s [1,2,3,4]. There is no sex difference in terms of the incidence or clinical and genetic features of GISTs, except GISTs with *SDH* alterations which appear to be relatively predominant in females. Multiplicity is rarely seen except among patients with familial predispositions for germline mutations in *KIT*, *PDGFRA*, or *SDH* [18,19,20] or for multiple small intestinal GISTs in neurofibromatosis type I patients [21,22] When patients have germline mutations in *KIT*, *PDGFRA*, or *NF1*, they may have early onset of GISTs, ICC hyperplasia in the normally appearing GI tract, and characteristic clinical features, such as skin pigmentation, dysphagia, and other tumors, in addition to multiple GISTs. When there are multiple GISTs in a patient without a hereditary background, we may consider that these tumors are multiple sporadic GISTs if each GIST has different *KIT* or *PDGFRA* mutations [23,24]. If they have the same mutation type, they may be considered a metastatic disease. There are no reported environmental risk factors for GISTs.

### 2.1. Pathological Diagnosis of GIST

The diagnosis of GISTs is based on pathological examinations, but not clinical examinations. Morphologically, GISTs can be divided into three types: the spindle cell type with eosinophilic fibrillary cytoplasm (70%), epithelioid type (20%) with clear eosinophilic cytoplasm, and mixed type with spindle and epithelioid cells (10%) [25,26,27]. Spindle cell-type GISTs should be differentiated from both benign and malignant diseases, including smooth muscle tumors (leiomyoma or leiomyosarcoma), schwannoma, hemangioma, plexiform fibromyxoma, desmoid, inflammatory myofibroblastic tumor (IMT), and solitary fibrous tumor (SFT), and epithelioid-type GISTs from melanoma, perivascular epithelioid cell tumor (PEComa), neuroendocrine tumors, clear cell sarcoma, and epithelioid variants of leiomyosarcoma [4,25,26]. Some characteristic pathological findings of each tumor are shown in Table 2. There are some correlations between clinicopathological features and the genotype of the GIST, as described later [28]. Epithelioid transformation or mixed type may also be found in aggressive GISTs in the small intestine.

Differentiation of GISTs from other tumors in the GI tract described above usually requires IHC in addition to hematoxylin and eosin staining, and, occasionally, genotyping (Table 2) [4,25,26,27]. In IHC, KIT (CD117) is expressed in ~95% of GISTs, and DOG1, a calcium-dependent, receptor-activated chloride channel protein, is expressed in ~95% of GISTs [25,26,27,28,29]. These two biomarkers usually show diffuse expression in tumor cells. KIT expression is regulated by ETV1, a transcription factor required for the proliferation of GISTs and ICCs, and the expression of which is, conversely, regulated by the MEK-MAPK pathway downstream of the KIT and PDGFRA tyrosine kinases [30]. It should be noted that melanoma, angiosarcoma, Ewing’s sarcoma, childhood neuroblastoma, seminoma, and small cell lung carcinoma may also show expression of the KIT protein by IHC [4,25,26,29]. In contrast, KIT expression is sometimes weak and faint in *PDGFRA*-mutated GISTs, in which DOG1 may be expressed [1,2,4,31]. CD34 may be expressed in GISTs but is less specific and less frequent (~70% of GISTs) [4,25]. S-100 is an immunohistochemical marker of neurogenic tumors, and alpha-smooth muscle actin and desmin are markers of myogenic tumors. As most GISTs with loss-of-function mutations in *SDH* subunits or with loss of expression due to methylation substantially do not express SDH subunit B, they are generally negative for SDHB in IHC [31]. A few GISTs may face diagnostic difficulty even with these IHCs and may require mutation research of the *KIT* and *PDGFRA* genes for their diagnosis.

### 2.2. Molecular Aspects of GIST

Molecularly, GISTs consist of heterogeneous subgroups, including GISTs with mutations in the *KIT*, *PDGFRA*, *SDH* genes, *RAS* genes, *BRAF*, *NF1,* or other rarely mutated genes as well as alterations [1,4,7,29,32,33]. KIT and PDGFRA have similar structures and similar downstream signaling pathways and are a type III receptor tyrosine kinase (RTK), a family including PDGFRB, CSF1R (macrophage colony-stimulating-factor receptor), and FLT3 (FMS-like tyrosine kinase 3) [34]. Small GISTs, including micro-GISTs and mini-GISTs, have *KIT* or *PDGFRA* mutations similar to those of clinical GISTs [16,35], and familial GISTs with germline mutations in *KIT* or *PDGFRA* accompanied by diffuse hyperplasia of ICC cells and multiple micro-GISTs and mini-GISTs with benign features [18,19,36]. These data and results from knocked-in mice indicate that mutations in the *KIT* or *PDGFRA* gene are an early neoplastic event and are considered to be causative of the GIST but are not always involved in malignant transformation [37].

The frequent driver mutations found in GISTs include mutations in *KIT* (~70%) or *PDGFRA* (10~15%), followed by mutations in *SDH* family genes, in *NF1*, in *BRAF*, in *RAS* family genes [1,4,7,29,32,33], or, rarely, in other gene alterations including fusion genes involving the TRK family [38,39]. The molecular subtypes may somewhat correlate with the primary location as well as clinicopathological features (Table 1). For example, *PDGFRA*-mutated GISTs, found mainly in the stomach, may frequently show epithelioid cell features, and *SDH*-GISTs are located mainly in the stomach and show epithelioid features separated by fibrous bands, whereas *NF1*-mutated GISTs usually appear as spindle cell tumors in the small intestine [19,20,21,22,25,40]. More importantly, mutations are considered the most reliable biomarker of medical therapy. The molecular correlation with the clinicopathological features of GISTs and drug sensitivities are briefly summarized in Table 1 and Figure 1. GISTs without *KIT* or *PDGFRA* mutations, so-called “Wild-type GISTs”, may be divided into SDHB-competent (SDHB-positive by IHC) and SDHB-deficient (SDHB-negative by IHC) GISTs [4,31]. The latter may have mutations in a subunit of the *SDH* complex, including *SDHA*, *SDHB*, *SDHC*, or *SDHD*, or may have downregulated expression of the SDH complex through site-specific hypermethylation of the promoter regions [9,19,20]. The *SDH*-GIST is resistant to all available tyrosine kinase inhibitors (TKIs) in most cases and may partly show transient stabilization or decrease in size under VEGFR inhibitor treatment because its progression is thought to be driven by the expression of insulin growth factor-1 receptor (IGF1R) and vascular endothelial growth factor receptor (VEGFR) induced by hypoxia-inducible factor-1α (HIF-1α) [4,40]. The former includes GISTs with mutations in *NF1*, *BRAF*, or *RAS*, which are usually accompanied by activation of the MEK-MAPK pathway, implying that GISTs with these mutations may be potentially sensitive to MEK inhibitors and/or BRAF inhibitors (Table 1) [41,42].

### 2.3. Clinical Diagnosis of GIST

GISTs are initially found as SMTs and/or abdominal masses during exploration of the GI tract due to the abovementioned symptoms and signs or are incidentally found during cancer screening as mentioned. Less frequently, a work-up for emergent admission with GI bleeding or perforation may reveal a GIST [4]. Clinical diagnosis is performed by endoscopy, endoscopic ultrasonography (EUS), ultrasonography and/or CT scan, and a definitive diagnosis can be made only by pathological examinations after surgery or biopsy sampling. Figure 2 shows the diagnostic flow of gastric SMT proposed in the Japanese GIST guidelines [43]. Although CT scans may have advantages in terms of diagnosis, especially for tumors showing extramural growth and for the evaluation of disease spread [44], CT scans have accompanying radiation exposure, and EUS is still a major diagnostic tool for GISTs in the stomach and rectum. Gastric GIST is frequently found by endoscopy and/or fluoroscopy; thus, the initial work-up with endoscopy and EUS is important in the differential diagnosis of GISTs from other neoplastic SMTs. Here, we quickly summarize the characteristic endoscopic and EUS features of several neoplasms found in the stomach (Table 2). The other important role of endoscopy and EUS may be to identify SMTs that require treatment, such as GISTs. An irregular shape (Figure 3a), ulcer formation (Figure 3b), and/or rapid growth between endoscopy intervals together with an irregular shape (Figure 3c), heterogeneous internal echo (Figure 3d), and/or regional lymph node swelling by EUS may indicate malignant tumors including a GIST, and, thus, these findings are considered as high-risk features for SMTs [2,12,45]. If patients with small SMTs have high-risk features, we recommend surgical resection or tissue acquisition for pathological diagnosis depending on tumor size, location, and conditions (Figure 2).

In practice, the pathological diagnosis of GISTs is infrequent before surgery [4], and the clinical diagnosis is not always consistent with the pathological diagnosis. The pathological diagnosis can be obtained by pre-treatment biopsy. Sampling biopsy is usually performed either endoscopically or percutaneously when neoadjuvant therapy is considered for locally advanced GIST or medical therapy for metastatic, recurrent, and/or unresectable GIST (hereafter “metastatic/recurrent GIST”). The biopsy method may be dependent on tumor location, disease spread, and accessibility.

### 2.4. Tissue Acquisition for Pathological Diagnosis

The diagnostic yield of conventional endoscopic forceps biopsy is low in GISTs and is accompanied by a relative risk of bleeding [46]. EUS-guided fine needle aspiration (EUS-FNA) is safe and useful for pathological diagnosis, although the rate of tissue acquisition of EUS-FNA for GISTs and SMTs varies dependent on the tumor, location, and skill of the specialist, and is lower than that for pancreatic lesions [46,47,48]. The diagnostic accuracy of EUS-FNA may be improved by the introduction of rapid on-site evaluation (ROSE) [49,50]. Recently, EUS-guided biopsy sampling (EUS-FNB), equipped with the side-fenestrated reverse bevel design needle, was shown to be more reliable in obtaining sufficient tissue (91%) from pancreatic cancer than EUS-FNA (67%) [51]. EUS-FNB may be useful for IHC and genomic sequencing and, thus, may be a promising theranostic of GISTs. Alternative endoscopic approaches may include mucosal incision-assisted biopsy (MIAB) using the technique of endoscopic submucosal dissection (ESD) or endoscopic mucosal resection (EMR). MIAB has been shown to have similar diagnostic accuracy and safety to EUS-FNA with ROSE [50].

Although EUS-guided biopsy is preferred to percutaneous biopsy in terms of the risk of tumor cell dissemination, a percutaneous biopsy may be required for small intestinal lesions and metastatic diseases, depending on the situation. Several retrospective studies have indicated that percutaneous biopsy does not increase the risk of recurrence among patients with localized high-risk GISTs in the settings of postoperative adjuvant therapy [52,53]. In clinical practice, when a patient presents with an abdominal tumor that is highly likely to be a GIST, and that is resectable without extended surgery, one may consider surgery (Figure 2 and Figure 4). In fact, when surgery is scheduled ahead, EUS-FNA is unlikely to be recommended in the clinical guidelines of the European Society of Gastrointestinal Endoscopy (ESEG) [54]. For metastatic/recurrent diseases, a percutaneous image-guided biopsy is feasible and appropriate. However, when a percutaneous biopsy is not applicable and/or accessible and when medical treatment is urgently required, imatinib may be started without biopsy after clinical diagnosis of GISTs. In these situations, we recommend early response evaluation by enhanced CT scan or PET-CT approximately 1 month after treatment.

## 3. Surgery

### 3.1. Surgical Therapy of Primary GISTs

Surgery is a mainstay and the only modality providing a permanent cure for primary localized GIST [2,4,43,55]. GISTs usually show expansive growth and rarely metastasize to lymph nodes except *SHD*-GIST. There is no known efficacy of prophylactic dissection of regional lymph nodes, and cherry-picking dissection of potentially metastatic lymph nodes is considered sufficient for GISTs, even for *SDH*-GISTs, which show frequent metastasis to lymph nodes [2,4]. GISTs are fragile and highly vascularized tumors; thus, they are gently manipulated and carefully de-vascularized during operations to avoid tumor rupture [2]. The principles of surgery in GIST cases include macroscopic complete resection (R0) and functional preservation of resected organs, ideally, by wedge resection [2,43]. R1 surgery may not always require re-excision in the imatinib era, and GISTs may be followed by watchful waiting with low relapsing risk, or may be treated with adjuvant therapy when the GIST is high risk in the risk stratifications and has imatinib-sensitive mutations [56,57,58].

For GISTs with imatinib-sensitive mutations, preoperative imatinib (neoadjuvant therapy) is recommended when the GIST is large, namely, when it is more than 10 cm, and/or is considered marginally resectable on technical grounds and location, or when the GIST is likely to have significant morbidity or functional deficit after surgery (Figure 4) [2,4,55,59,60]. The preoperative treatment period may be between approximately 6 and 12 months and should not exceed 1 year. Early evaluation of imatinib activities approximately 1 month after treatment is important, and imatinib could be continued when there is no disease progression by enhanced CT scan. When imatinib treatment is active, GISTs are decreased not only in size but also in vascularity, which may increase the safety of surgery and may prevent intraoperative rupture [61,62]. In a few cases, however, massive necrosis of GIST tumor cells may cause inflammatory responses and fibrous adhesions to surrounding tissues, and GIST may become brittle. Imatinib neoadjuvant therapy shows significant safety and feasibility [61,63]. The efficacy in prognostic improvement, preservation of organ function, such as in duodenal GISTs and rectal GISTs, and increase in R0 resection, as well as resectability, has yet to be determined, although several retrospective and prospective studies have indicated these possibilities [61,62,63,64]. At present, most GISTs undergoing neoadjuvant therapy are considered to be high-risk GISTs, at least before the treatment, and may be recommended to receive adjuvant therapy even after R0 surgery.

### 3.2. Surgical Therapy of Small GISTs

The incidence of clinical GISTs is estimated to be 10/million/year, whereas that of mini-GISTs is reportedly 1/1000, as described above. Several retrospective cohort studies have shown that there are a small but significant number of recurrences (less than 10%) after R0 surgery of GISTs less than 2 cm (<2 cm GIST) after 10 years of follow-up [65,66,67]. A subanalysis of the large epidemiologic study of <2 cm GISTs, however, did not show a significant decrease in disease-specific mortality of patients by surgical resection compared with observation (10.9% vs. 27.9%, *p* = 0.13), probably due to low statistical power [67]. Gastric GISTs have different immunohistochemical and genetic features from small intestinal GISTs [68,69]. The former is thought to be clinically indolent compared with the latter [2,4,11,24,65]. Most gastric mini-GISTs do not show malignant features or behaviors [2,12,13,16]; thus, they may not become clinical GISTs. There is inconsistency in the recommendations for surgical resection of gastric GISTs <2 cm. Surgical resection is recommended for gastric GISTs <2 cm in the Japanese and Asian GIST guidelines (Figure 2; upper panel) [43,55], whereas the NCCN guidelines recommend surgical resection for gastric GISTs <2 cm when they have “high-risk features” based on empirical evidence; otherwise, they could be followed by periodical EUS after shared decision-making (Figure 2; lower panel) [2]. All guidelines, including the NCCN and ESMO guidelines [2,43,55,59], recommend complete resection for GISTs in the rectum that are <2 cm because of their different clinical features and prognostic outcomes.

There have been many reports describing endoscopic resection of small GISTs and SMTs using ESD techniques, endoscopic full-thickness resection (EFTR), or others [70,71,72,73]. Endoscopic resection of small GISTs has been shown to be safe and feasible, and it has shown good prognostic outcomes. However, we need to be careful in interpreting these prognostic outcomes. Most gastric mini-GISTs may have an indolent clinical course and do not progress even without resection [12,16,74,75]. Furthermore, GIST patients have significant competing risks, including secondary cancer and cardiovascular diseases [57,76]. Hence, we need to identify features of small gastric GISTs indicating disease progression, resulting in a poor prognosis without treatment. Thus, it is necessary to narrow down small gastric GISTs requiring surgical resection. In this regard, the clinical features of GISTs indicating highly malignant potential are indicated to include an irregular shape, mucosal ulceration, and tumor size > 2 cm [77]. High-risk features of small gastric GISTs should be re-evaluated by prospective studies, and indications of surgery for gastric GISTs <2 cm should be reconsidered.

### 3.3. Laparoscopic Surgery for GISTs

Laparotomy has been standard in GIST surgery; now, laparoscopic surgery is also considered the standard procedure for surgery in cases of small GISTs less than 5 cm [4,12]. Laparoscopic surgery for GISTs has been shown to be less invasive, less painful, and have a faster postoperative recovery and better cosmetic outcomes than open surgery, with similar oncologic prognoses [12,78]. In laparoscopy, concomitant use of endoscopy may facilitate securing oncological margins and adequate patency of the remnant gastrointestinal lumen. One of the typical surgeries includes laparoscopic endoscopic cooperative surgery (LECS) and related procedures, which are feasible and safe in short-term outcomes and have similar oncological outcomes to those of open surgery after long-term follow-up [79,80]. These procedures benefit organ function preservation by minimizing resection of normal organs in addition to being less invasive. They may work for surgery of GISTs near the esophagogastric junction (EGJ) or pylorus, although surgery itself is technically demanding in these locations.

Initially, laparoscopic surgery is predominantly performed for GISTs that are smaller than 5 cm, but currently, it is applied to GISTs larger than 5 cm. Several retrospective studies and their meta-analyses suggest that laparoscopic surgery for large GISTs shows similar operation times, and less blood loss, postoperative morbidity, and shorter in-hospital days than open surgery [81,82,83]. Long-term oncologic outcomes in terms of disease-free survival (DFS) and overall survival (OS) are similar between the two. However, evidence of laparoscopy is very limited when the tumor is over 8 cm, and the application of laparoscopy may vary depending on the tumor location and conditions.

### 3.4. Risk Evaluation in GIST

Risk assessment in localized GISTs aims to identify GISTs likely to recur after surgery, hence, to identify GISTs requiring multidisciplinary treatment and/or intense follow-up. Tumor size, location, and mitotic count of tumor cells under a microscope are well-established independent prognostic factors [2,27,84]. They are included in several risk stratifications and nomograms, such as the National Institutes of Health (NIH) consensus criteria, the Armed Forces Institute of Pathology (AFIP) criteria, the modified NIH classification, and the Gold’s nomogram [84,85,86,87]. In the stratification systems, size was categorized as <2 cm, 2~5 cm, 5~10 cm, and >10 cm, and mitosis was categorized as <5/5 mm^2^, 5~10/5 mm^2^, > 10/5 mm^2^. However, these factors are continuous variables showing a non-linear relationship with recurrence risk [65]. Afterward, the prognostic contour map and the Gold’s nomogram were introduced [84,87]. Genotype, the presence of clinical symptoms, and histological necrosis, among other factors, have been reported to be possible prognostic factors; however, no factor is superior to size, location, or mitosis as an independent prognostic factor [4,88]. For example, GISTs with deletion mutations of codons 557–558 have been shown to have aggressive clinicopathological features and poorer prognosis, and most *PDGFRA*-mutated GISTs and *SDH*-GISTs show indolent features and better prognoses; however, they are not always shown to be independent for prognostic evaluation.

The modified NIH classification includes tumor rupture as a prognostic factor [65,84]. Rupture is a clinical factor that was not universally defined. Some retrospective studies have reported rupture as an independent prognostic factor, whereas others have not, probably due to different criteria for tumor rupture [58,89,90,91]. Recently, the universal definition of tumor rupture was proposed [58,89]. The composite definition of the rupture includes tumor fracture or spillage, blood-stained ascites, gastrointestinal perforation at the tumor site, microscopic infiltration of an adjacent organ, piecemeal resection, or incisional biopsy. In contrast, R1 surgery, intraluminal penetration of the tumor, needle biopsy, and peritoneal penetration of tumor cells in pathological examinations were not considered tumor rupture. Even with this definition, 10 to 20% of GISTs with tumor rupture do not recur during follow-up; in particular, GISTs with low mitotic counts show low recurrence rates even in the presence of tumor rupture [92]. The prognostic significance of tumor rupture should be prospectively re-evaluated according to the definition.

When patients have a significant risk of recurrence, imatinib adjuvant therapy is indicated after R0/R1 surgery. Details of the indication and duration of adjuvant therapy are discussed elsewhere [2,4,11,59,84], and, here, we briefly provide an overview of adjuvant therapy. Values indicating “significant risk” may vary depending on individuals. Clinical studies show that patients with high-risk GISTs in the risk stratifications may benefit from adjuvant therapy. Recurrence rates of high-risk GISTs may be estimated to be more than 40~50% after 10 years of follow-up [65,84]. The other important factor to be considered is imatinib sensitivity [4]. The guidelines do not recommend adjuvant therapy for *PDGFRA* D842V-mutated GISTs [2,4,59], nor is the therapy indicated for GISTs without *KIT* or *PDGFRA* mutations (“wild-type GIST”) because of the relatively indolent nature as well as their lack of imatinib responsiveness [59]. Clinical evidence suggests that adjuvant therapy for 3 years improves recurrence-free survival (RFS) as well as OS among patients with high-risk GISTs compared with 1-year adjuvant therapy [92,93]. The duration of adjuvant therapy may depend on the estimated recurrence risk and patient preference as well as conditions [4,59] and has not yet been established. Five years of adjuvant therapy shows that recurrences are rare during therapy and are frequently observed within a couple of years after stopping imatinib [94]. Recurrence after discontinuation of adjuvant therapy is very similar among 1-year, 2-year, 3-year [95], and 5-year adjuvant therapies, suggesting that imatinib activities are cytostatic. Thus, we may consider longer adjuvant therapy for very high-risk GISTs, such as ruptured GISTs. In fact, the ESMO guidelines indicate life-long adjuvant therapy for ruptured GISTs [59], if tolerable and if GISTs have imatinib-sensitive mutations.

## 4. Medical Therapy

### 4.1. Medical Therapy for Metastatic/Recurrent GISTs

TKIs are the primary choice for metastatic/recurrent GISTs [2,3,4,11,59]. Currently, five TKIs, namely, imatinib, sunitinib, regorafenib, ripretinib, and avapritinib, have significant clinical evidence for GIST treatment, but insurance reimbursement for recent two TKIs, ripretinib and avapritinib, may depend on the country. The details of the initial three TKIs, namely, imatinib, sunitinib, and regorafenib, are not addressed in this manuscript and are referred elsewhere for this information [2,3,4,11]. This paper focuses on emerging therapy and newly developing drugs for GISTs. Briefly, metastatic/recurrent GISTs with conventional *KIT* or *PDGFRA* mutations stabilized in the autoinhibited form (Table 1 and Figure 1) are initially treated with imatinib, and when resistance or intolerance to imatinib develops, sunitinib serves as the second-line treatment. When the GIST becomes refractory to sunitinib, regorafenib is used as the third-line, followed by ripretinib as the fourth-line treatment (Figure 1 and Table 1) [4,96]. When *PDGFRA* D842V mutations stabilized in the activated form are found, GISTs may be treated with avapritinib [4,97]. When there is no mutation in either the *KIT* or *PDGFRA* gene, the GIST should be subjected to targeted gene panel analysis or whole-exome sequencing; then, the patient can be advised to receive potential therapeutic agents based on the results (Figure 1 and Table 1). When no mutational information is available, the conventional first-line imatinib, second-line sunitinib, and third-line regorafenib are recommended. Regarding the correlation between mutation types and TKI activity, sunitinib has significant activity against *KIT* exon 9-mutant GISTs after imatinib therapy [98]. The majority of GISTs, which are initially responsive to imatinib, become refractory to the drug due to secondary mutations in either the ATP-binding domain (exons 13 and 14) or the activation loop domain (exons 16, 17, and 18) [99]. Sunitinib is active for ATP-binding domain mutations, but not for mutations in the activation loop [100]. Regorafenib, in contrast, has significant activity against GISTs with mutations in the activation loop [101,102].

### 4.2. Newly Emerging Therapy: New TKIs and Drugs for the NTRK Fusion

Ripretinib, a switch-control TKI inhibiting both *KIT* and *PDGFRA* kinases by securing the kinases in an inactive conformation, inhibits most primary and secondary mutations of *KIT* and *PDGFRA* in vitro [103]. In a pivotal phase III study, ripretinib was compared with placebo in GIST patients previously treated with imatinib, sunitinib, and regorafenib [96]. Progression-free survival (PFS; median PFS: 6.3 months for ripretinib and 1.0 months for placebo; hazard ratio (HR) = 0.15, *p* < 0.0001) and OS (median OS: 15.1 months for ripretinib and 6.6 months for placebo; HR = 0.36; *p* = 0.0004) were better with ripretinib. Toxicity was mild, and the drug was well tolerated. The most common adverse events observed in the ripretinib arm were alopecia, nausea, diarrhea, myalgia, fatigue, and palmar-plantar erythrodysesthesia syndrome. The FDA approved ripretinib for patients with metastatic/recurrent GIST who have received prior treatment with three or more kinase inhibitors. A phase III study comparing ripretinib with sunitinib in the second-line therapy for metastatic/recurrent GISTs is currently in progress.

Avapritinib is another oral TKI designed to selectively target the active conformation of KIT and PDGFRA via a type 1 inhibition mechanism and inhibits various *KIT* and *PDGFRA* mutations, including those resistant to the three approved TKIs. In the phase I study, avapritinib showed substantial clinical activity against *PDGFRA*-mutant GISTs [97]. Among patients with *PDGFRA* D842V-mutant GISTs, the response rate was 88% (49 of 56 patients), with five (9%) complete responses and 44 (79%) partial responses. There were no dose-limiting toxicities at doses of 30–400 mg per day. The FDA has approved avapritinib for *PDGFRA* exon 18 mutation-positive unresectable or metastatic GISTs. Avapritinib was evaluated in the third- or fourth-line settings by comparison with regorafenib, and failed to demonstrate superiority to regorafenib in terms of PFS (median PFS 4.2 months for avapritinib and 5.6 months for regorafenib; HR = 1.25; *p* = 0.055) [104].

GISTs without *KIT* and *PDGFRA* mutations may be called “wild-type GISTs”. A small proportion of gastrointestinal mesenchymal tumors and “wild-type GISTs” may have NTRK fusions, which are candidates for TRK inhibitors [105], although there is some discussion on GISTs with *NTRK* fusions. It is reported that *NTRK* rearrangement-containing mesenchymal tumors in the GI are clinically and morphologically heterogeneous, and that few may be related to GISTs [39]. Even so, larotrectinib, a selective oral TRK inhibitor, the first drug approved for solid tumors with an *NTRK* gene fusion, has shown a response rate (RR) of 79% in the pooled analysis of a phase 1 study in adults, a phase 1/2 study in children, and a phase 2 basket study [106]. Common adverse events include fatigue, dizziness, nausea, vomiting, increased AST, and cough. Entrectinib, which inhibits ROS1, ALK, three TRKs, and TRK-fusion tyrosine kinases, has shown an RR of 57% (31 of 54 patients) [107], and is approved by the FDA for *NTRK* gene fusion-positive solid tumors and ROS1-positive non-small cell lung cancer.

### 4.3. Developing Therapy

There are many developing drugs for GISTs, and here, we quickly discuss a few emerging drugs other than TKIs. Immune checkpoint inhibitors have been approved for a wide range of tumors, including lung cancer, melanoma, head and neck cancer, esophageal cancer, gastric cancer, urothelial cancer, and breast cancer, among others. Anti-programmed death-1 (PD-1) and anti-programmed death-ligand 1 (PD-L1) antibodies have also been evaluated in GISTs. A randomized phase II trial including 40 metastatic/recurrent GIST patients explored nivolumab, a monoclonal antibody for PD-1, combined with or without ipilimumab, a monoclonal antibody for the human T-cell receptor cytotoxic T-lymphocyte-associated antigen 4 (CTLA4), and the results showed only one response among 12 patients in the combination arm [108]. Pembrolizumab, an anti-PD-L1 antibody, was evaluated with epacadostat, a selective indoleamine 2,3-dioxygenase (IDO1) inhibitor; however, the study was terminated earlier due to insufficient clinical efficacy [109].

HSP90 is a molecular chaperone that stabilizes client proteins, including KIT, BRAF, and SDHs. HSP90 inhibitors have been evaluated in TKI-resistant GISTs. However, the development of a first-generation of HSP90 inhibitors, such as retaspimycin (IPI-504) and luminespib (AUY922), has failed to develop due to drug toxicities [110,111]. Additionally, the second-generation of pimitespib (TAS-116), an oral competitive inhibitor of cytosolic Hsp90α and β, showed a meaningful disease control rate of 85.0% and a median PFS of 4.4 months in the phase II study (n = 40). The drug was compared with placebo in a phase III clinical study. The results showed significantly improved PFS (median PFS = 2.8 months for pimitespib and 1.4 months for placebo; HR = 0.51; *p* = 0.006), and the median OS times were 13.8 months for pimitespib and 9.6 months (HR = 0.63; *p* = 0.081) for placebo, with tolerable safety profiles [112,113].

Selinexor, an oral selective inhibiter of nuclear export that functions by blocking exportin 1 (XPO1), is currently being explored in combination with imatinib for GISTs [114]. DS-6157a is an antibody-drug conjugate targeting G protein-coupled receptor 20 (GPR20), which is selectively expressed in GISTs [115], and currently, a phase I study is underway in the US and Japan. XmAb18087, a bispecific antibody against somatostatin receptor 2 (SSTR2) and CD3, is now being explored in a phase I study of patients with metastatic/recurrent GISTs [116].

## 5. Conclusions

GISTs are potentially malignant tumors in the GI tract, and clinical GISTs require treatment; however, the treatment impact for gastric GISTs that are less than 2 cm may be reconsidered because of their indolent nature and competing risks among these patients. Preoperative diagnosis of GISTs is challenging, and a work-up by means of endoscopy and EUS as well as endoscopic or percutaneous biopsy may be useful for the differential diagnosis and subsequent therapy of GISTs. Laparoscopy has similar safety and prognostic outcomes to those of laparotomy in terms of surgery of GISTs, including for tumors larger than 5 cm. Medical treatment with TKIs is the mainstay for recurrent/metastatic GISTs. The activity of each drug is well correlated with gene mutations and alterations; thus, in the era of precision medicine, cancer genome profiling should be considered when treatments are used. Targeted gene panel analysis and whole-exome sequencing may provide potential targeted therapy for “wild-type GISTs” and GISTs that are refractory to conventional TKIs.

## Authors Contribution

Conceptualization, T.N.; Writing, T.N., S.Y., T.T. and Y.N.; Visualization, T.N., S.Y. and T.T.; Funding acquisition, T.N. All authors have read and agreed to the final version of the manuscript.

## Figures and Tables

**Figure 1 cancers-13-03158-f001:**
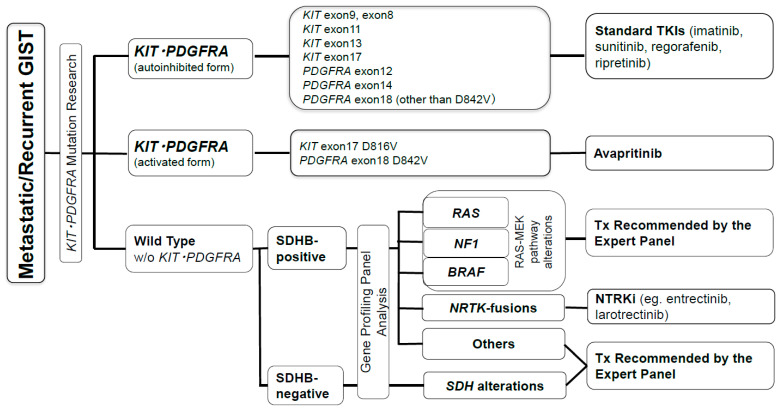
Mutational types and medical treatment for metastatic/recurrent GISTs. Abbreviations: TKI: tyrosine kinase inhibitor, Tx: therapy, NTRKi: NRTK inhibitor.

**Figure 2 cancers-13-03158-f002:**
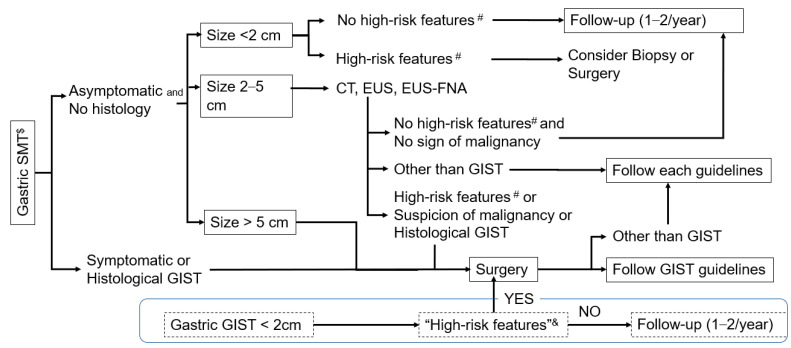
Diagnostic and treatment flow of gastric submucosal tumors and GISTs. Based on the Japanese GIST guidelines, the diagnostic and treatment flow of gastric submucosal tumors, including GISTs, is shown in the upper panel. In the lower panel, the approach to gastric GISTs < 2 cm is indicated according to the NCCN guidelines. $: SMT: submucosal tumor, #: high-risk features: irregular shape, ulcer formation, and/or rapid growth between intervals by endoscopy, and heterogeneous internal echo, irregular shape, and/or regional lymph node swelling by EUS (Ref. [43]), &: “high-risk features” including irregular border, cystic spaces, ulceration, echogenic foci, and heterogeneity (Ref. [2]). The former high-risk features (#) used in the Japanese GIST guidelines are endoscopic and EUS features of submucosal tumors that require treatment, whereas the latter “high-risk features” (&) in the NCCN guidelines include EUS features suggesting small gastric GISTs with potential disease progression.

**Figure 3 cancers-13-03158-f003:**
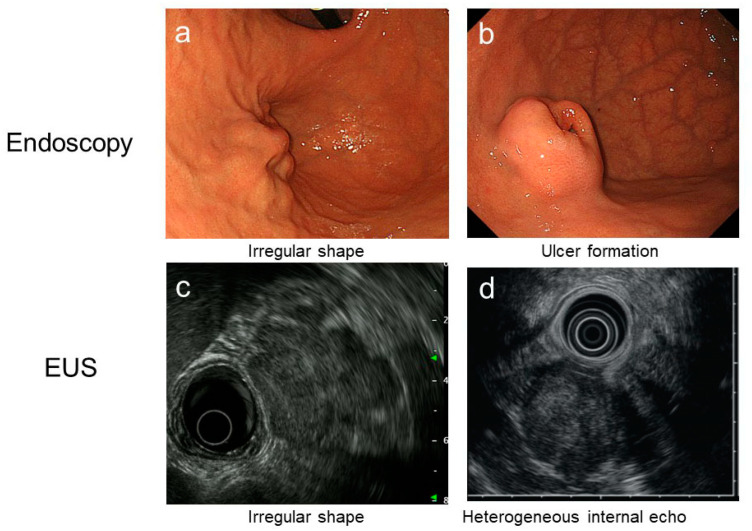
High-risk features by endoscopy and EUS. Representatives of high-risk features by endoscopy and EUS are shown.(**a**): endoscopic features of GIST with irregular shape; (**b**): endoscopic features of GIST with ulcer; (**c**): EUS features of GIST with irregular shape; (**d**): hypoechoic EUS features of GIST with heterogeneous internal echo.

**Figure 4 cancers-13-03158-f004:**
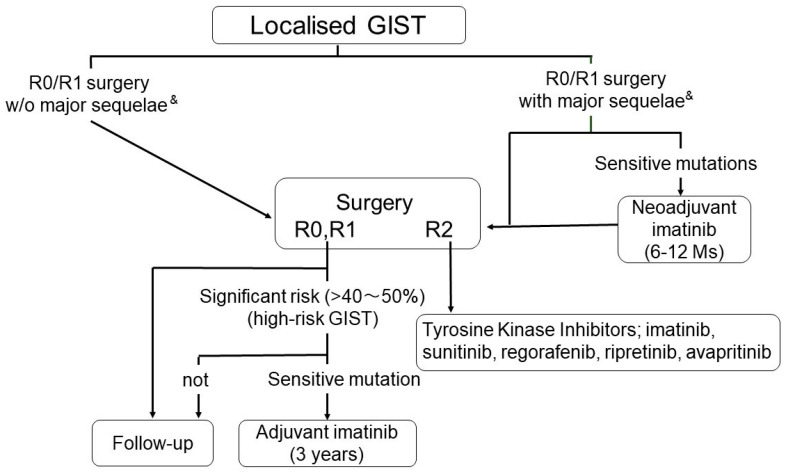
Management of localized GISTs. &: “R0/R1 surgery with major sequela” indicates that surgery could reasonably be considered to be accompanied with risk for postoperative major morbidities and/or functional deficit.

**Table 1 cancers-13-03158-t001:** Features and mutations of GISTs.

		Alterations	Estimated Frequency	Main Location	Characteristics	Sensitivity to Drugs & Potential Drugs	
*KIT* mutations in the autoinhibited form	*KIT* mutation #	exon 9 (or exon 8), typically duplicated insertion of A502-Y503 codons	5–10%	Small intestine	Spindle cell type Aggressive features	Less imatinib sensitive, sensitive to sunitinib, regorafenib, ripretinib, avapritinib	
		exon 11 (deletions, missense, insertions etc.)	~60%	All sites	Aggressive features with del 557-558, which is very sensitive to imatinib	Sensitive to imatinib, sunitinib, regorafenib, ripretinib, avapritinib	
		exon 13 (K642E)	<1%			Sensitive to imatinib, sunitinib, regorafenib, ripretinib, avapritinib	
		exon 17 (D820Y, N822K, Y823D)	1%			Sensitive to imatinib, regorafenib, ripretinib, avapritinib, and less sensitive to sunitinib	
*PDGFRA* mutations in the autoinhibited form	*PDGFRA* mutation #	exon 12 (V561D etc.)	<1%	Stomach>>small intestine	Epithelioid cell type Indolent clinical course in main	Probably sensitive to imatinib, sunitinib, regorafenib, ripretinib, avapritinib	
		exon 14 (N659K)	<1%			Probably sensitive to imatinib, sunitinib, regorafenib, ripretinib, avapritinib	
		exon 18 (del, Y849H etc., other than D842V)	1–2%			Sensitive to imatinib, sunitinib, regorafenib, ripretinib, avapritinib	
*KIT* or *PDGFRA* mutations in the activated form		*PDGFRA* exon 18 D842V, rarely *KIT* exon 17 D816V	~10%	Stomach>>small intestine	Epithelioid cell type	D842V is resistant to imatinib, sunitinib, regorafenib. D842V is sensitive to avapritinib & ripretinib	
No mutation in *KIT* and *PDGFRA*	SDHB-competent	*NF1* mutation $	1–2%	Small intestine	Spindle cell type Generally indolent clinical course associated with Neurofibromatosis type I	not sensitive to available drugs	possibly sensitive to MEK inhibitors, such as selumetinib
		*BRAF* mutation	<1%	Small intestine/stomach	Spindle cell type VE1-positive		possibly sensitive to BRAF inhibitors (e.g., vemurafenib, dabrafenib)
		*HRAS, NRAS or KRAS* mutation	very rare	no data	no data		MEK inhibitors (e.g., trametinib) may possibly have some activities
		Others including *PIK3CA*, *CBL*, *ETV6–NTRK3* et al.	very rare	no data	no data		NTR-fusion is sensitive to entrectinib and larotrectinib
	SDHB-deficient	*SDHA, SDHB, SDHC* or *SDHD* mutation (including Carney-Stratakis syndrome #)	~3%	Stomach>>small intestine	Epithelioid cell type SDHB-negative Children/adolescent and young adult Frequent lymph node metastasis Indolent clinical course	not sensitive to available drugs VEGFR inhibitors may have temporary stabilizing effects	
		Loss of SDHB expression (including Carney Triad $)	<1%	Stomach			

#: there are some GISTs with germline mutations; familial GIST; $: syndromic GIST.

**Table 2 cancers-13-03158-t002:** Endoscopic and EUS features of gastric submucosal tumor.

Disease	Endoscopic Findings		EUS Findings		Pathological Features	
	Surface, Form, etc.	Major Location	Main Layer	Echo Findings	Morphology	IHC; Genetic Changes
GIST	hemi-spherical, occasionally with delle or ulcer	body	proper muscle, rarely submucosa	hypoechoic, heterogenous with increased malinancy	spindle cell > epithelioid cell	KIT, DOG1; mutation in *KIT* or *PDGFRA*
Myogenic tumor & Leiomyoma	hemi-spherical, intact mucosa	near cardia	proper muscle, sometimes submucosa	round, hypoechoic, homogenous	spindle cell (eosinophilic cell)	Desmin, α-SMA
Schwanomma & neurogenic tumor	hemi-spherical, intact mucosa	body, lesser curvature	proper muscle, sometimes submucosa~deep mucosa	hypoechoic, homogenous~slightly heterogeneous	spindle cell, palisading, Verocay body, lymphoid cuff in Schwannoma	S-100, SOX10, NSE in neurogenic tumor
Heterotopic Pancreas	hill-shaped, intact mucosa, maybe dimple or aperture	antrum	submucosa	sometimes lobulated, ductal structure, heterogeneous internal echo, thickend proper muscle	Heimlich classification ^&^	
Neuroendocrine tumor	hemi-spherical, mucosal color~yellowish~red, occasionally dimple	body	initially deep mucosa or submucosa	homogenous, heterogeneous with increased malinancy	epithelioid cell, organoid pattern	CD56, synaptophysin, chromogranin A, NSE
MALT lymphoma	various surface, multiple lesions	anywhere	deep mucosa~submucosa	beltlike~multiple round, hypoechoic, homogenous	Centrocyte-like, lymphoepithelial lesion, plasma cell differentiation	κ or λ chain; t(11;18)/*API2-MALT1*
Malignant lymphoma	various surface, multiple lesions	anywhere	initially deep mucosa~submucosa	beltlike~advanced carcinoma-like, hypoechoic, homogenous		CD20+, CD79a+; t(3;14)/*BCL6-IGH*
Lipoma & lipogenic tumor	hill-shaped to pedunculated, intact mucosa (~yellowish), cushion sign	antrum	submucosa	round~oval, hyperechoic	Lipoblast (spider-web cell)	MDM2, CDK4 in well differenciated liposarcoma
Granular cell tumor	hemi-spherical, molar-like appearance, intact~ivoly	body	submucosa	round, heterogenously hypoechoic	eosinophilic granules	S-100, SOX10, CD68
inflammatory fibroid polyp (IFP)	pedunculated or penis-like, may with erosion/ulcer	antrum	deep mucosa~submucosa	hypoechoic, relatively homogeneous	perivascular fibrosis (onion skin pattern), eosinophil infiltration	CD34, α-SMA; mutations in *PDGFRA*
inflammatory myofibroblastic tumor (IMT)	hill-shaped, mucosal color	fornix~body		hypoechoic (not definite)	spindle cell & inflammatory cell infiltration	ALK, α-SMA, ALK-fusion, CD34,
Solitary fibrous tumor (SFT)	n.d.	n.d.		n.d.	spindle cell, patternless pattern	CD34, nuclear STAT6, bcl2, CD99; *NAB2-STAT6* fusion
Glomus tumor	hemi-spherical, same color as mucosa	antrum	proper muscle	relatively hyperechoic~heterogenous	eosinophilic cell with oval nucleus	α-SMA
lymphangioma or cavenous hemangioma	flat-elavated, intact mucosa (whitish or dark-reddish, respectively), cushion sign	n.d.	deep mucosa~submucosa	aechoic~hyperechoic, multicystic	endothelial cells	CD31, CD34, Factor VIII in vascular tumor
PEComa	hemi-spherical~polypoid, intact mucosa	n.d.	submucosa	hypoechoic, homogenous	epithelioid cell with clear cytoplasm	α-SMA, HMB45, Melan A; LOH of *TSC2*
Melanoma	pedunculated or lobular protrusion (may with melanosis), occasionally with erosion	n.d.	initially deep mucosa~submucosa	iso-hypoechoic, maybe regional lymph node metastasis		S-100, HMB45, Melan A, SOX10; mutations in *BRAF* ot *KIT*
Desmoid	n.d.	n.d.		n.d.	spindle cell	nuclear β-catenin; alterations in *CTNNB1*
Metastatic tumor	bull’s eye~various, multiple lesions	n.d.	deep mucosa~submucosa	round~oval, hyperechoic & heterogenous		

n.d.: no definite data; &: Heimlich classification; Type 1: normal pancreatic tissue, Type 2: acinar cells & ducts, Type 3: ducts & hyperplasia of smooth muscle fibers.

## Data Availability

The data presented in this study are available in this article.

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
