# Peer review of "Recent Progress and Challenges in the Diagnosis and Treatment of Gastrointestinal Stromal Tumors"

_cancers, 2021, doi:10.3390/cancers13133158_

Round 1
Reviewer 1 Report
This manuscript entitled by “Recent Progress and Challenges in the Diagnosis and Treatment of Gastrointestinal Stromal Tumors” by Toshirou Nishida comprehensively reviewed the current development of GIST from diagnosis, local treatment and systemic treatment. Generally, I consider this manuscript is well-prepared and well-written and suitable for publication after some revisions.
- English writing is good but still some errors were found such as
- within 60 and 69 years à 60 to 69 years
- Line 396: avaprinitib à avapritinib
- The format of citation (lines 106, 112: 4,25,26)
- The texts in tables 1 and 2 are too small
- Authors may clarify “R0/R1 surgery with major sequela” and “w/o major sequela” in figure 4. I proposed “R0/R1 surgery with major sequela” and “w/o major sequela” have been done based on the flow-chart when I firstly viewed this figure. However, after I reviewed this figure for longer time, I realized this should the proposed outcomes. Therefore, authors should clarify “R0/R1 surgery with major sequela” and “w/o major sequela” in figure 4. .
- Line 257, in terms of preoperative imatinib (neoadjuvant therapy), the authors may cite some clinical references other than guideline, such as doi: 10.3390/cancers11030424.
- Line 272, At present, most GISTs undergoing neoadjuvant therapy are considered to be high-risk GISTs, at least, before the treatment, and may be recommended to receive adjuvant therapy even after R0 surgery. This concept cannot be found in the figure 4.
Author Response
Reply from the authors to comments from the reviewers
Reviewer 1
This manuscript entitled by “Recent Progress and Challenges in the Diagnosis and Treatment of Gastrointestinal Stromal Tumors” by Toshirou Nishida comprehensively reviewed the current development of GIST from diagnosis, local treatment and systemic treatment. Generally, I consider this manuscript is well-prepared and well-written and suitable for publication after some revisions.
Thank you your kind reviewing our MS. As indicated, we’ve made pint-by-point reply as follows:
- English writing is good but still some errors were found such as
within 60 and 69 years à 60 to 69 years
Line 396: avaprinitib à avapritinib
Reply:
Thank you for your kind indications. We’ve corrected each as follows:
the reported median ages are 60s years
“avaprinitib” is changed to “avapritinib”.
- The format of citation (lines 106, 112: 4,25,26)
Reply:
Thank you for your kind indications, but, this may be due to troubles in transformation in WEB, because they works as 4,25,26 in the original word file. Now, they are fixed.
- The texts in tables 1 and 2 are too small
Reply:
We are very sorry for your inconvenience. We’ve improved by dividing tables 1 and 2 into two parts.
- Authors may clarify “R0/R1 surgery with major sequela” and “w/o major sequela” in figure 4. I proposed “R0/R1 surgery with major sequela” and “w/o major sequela” have been done based on the flow-chart when I firstly viewed this figure. However, after I reviewed this figure for longer time, I realized this should the proposed outcomes. Therefore, authors should clarify “R0/R1 surgery with major sequela” and “w/o major sequela” in figure 4. .
Reply:
We appreciate the reviewer’s suggestion. We intended that “R0/R1 surgery with major sequela” might mean that R0/R1 surgery could reasonably be considered to have risk for postoperative major morbidities or functional deficit. So, we added the comment of “# R0/R1 surgery with major sequela: surgery could reasonably be considered to be accompanied with risk for postoperative major morbidities or functional deficit” in the footnote.
- Line 257, in terms of preoperative imatinib (neoadjuvant therapy), the authors may cite some clinical references other than guideline, such as doi: 10.3390/cancers11030424.
Reply:
We appreciate the reviewer’s suggestion. We have cited some references of Ref. 2, 4, 54, and 58 in the very last of sentence. We’ve cited your suggestion as Ref. 60 (because of another new citation in Ref. 27), accordingly, Ref. Numbers after 60 are changed.
- Line 272, At present, most GISTs undergoing neoadjuvant therapy are considered to be high-risk GISTs, at least, before the treatment, and may be recommended to receive adjuvant therapy even after R0 surgery. This concept cannot be found in the figure 4.
Reply:
Thank for your indication. We consider that patients undergone neoadjuvant therapy due to huge size and/or risks of postoperative major morbidities and/or functional deficit had better to have adjuvant therapy after surgery. These are just our idea and suggestions, and no solid clinical evidence. So, we do not incorporate it into the flowchart of Fig. 4, and it is just described in the text.

Reviewer 2 Report
Nishida et al. present a review of GIST with a specific focus on diagnosis and treatment. While the topic is quite broad, they summarize some of the most relevant studies in the field providing an interesting and updated overview, despite some issues with the English language that requires some revision. A few additional points below:
-I do not see any reference to the last WHO classification of tumors-5th edition-Soft Tissue and Bone Tumors published in 2020
-Tables 1 and 2: character is too small, especially in table 2 and makes it difficult to read
-the term “theranostic” that is uses at least a couple of times such as in line 40 (abstract) does not seem appropriately used; I would just avoid it
-sometimes references are not in upper cases (ex: line 112)
-lines 157-158: the sentence “SDH-GIST is resistant to all available TKIs…” is not totally true given some responsiveness to sunitinib or regorafenib, despite refractoriness to imatinib
-lines 213-214: can you roughly quantify the risk of bleeding?
-line 329: local expertise also dictates the application of laparoscopy
-line 390 and 488: “medicines” should be replaced with other word such as “treatments”
-Figure 1: the word “Avapritib” is misspelled
Author Response
Reply from the authors to comments from the reviewers
Reviewer 2
Nishida et al. present a review of GIST with a specific focus on diagnosis and treatment. While the topic is quite broad, they summarize some of the most relevant studies in the field providing an interesting and updated overview, despite some issues with the English language that requires some revision. A few additional points below:
Thank you your kind reviewing our MS. As indicated, we’ve made pint-by-point reply as follows:
-I do not see any reference to the last WHO classification of tumors-5th edition-Soft Tissue and Bone Tumors published in 2020
Reply:
We appreciate reviewer’s suggestion, and the following reference is added in the Pathological Diagnosis of GIST section as Ref. 27.
Dei Tos AP, Hornick JL, Miettinen M, Wanless IR, Wardelmann E. Gastrointestinal Stromal Tumour. In: The WHO Classification of Tumours Editorial Board, eds. WHO Classification of Tumours Soft Tissue and Bone Tumours, 5th ed. Lyon: IARC Press; 2020:pp216–221.
-Tables 1 and 2: character is too small, especially in table 2 and makes it difficult to read
Reply:
We are very sorry for your inconvenience. We’ve improved by dividing tables 1 and 2 into two parts.
-the term “theranostic” that is uses at least a couple of times such as in line 40 (abstract) does not seem appropriately used; I would just avoid it
Reply:
We appreciated suggestion and we’ve corrected just to “useful for diagnosis” in the abstract. The other is left as it is used in the original paper.
-sometimes references are not in upper cases (ex: line 112)
Reply:
Thank you for your kind indications, but, this may be due to troubles in transformation in WEB, because they works as4,25,26 in the original word file. Now, they are fixed.
-lines 157-158: the sentence “SDH-GIST is resistant to all available TKIs…” is not totally true given some responsiveness to sunitinib or regorafenib, despite refractoriness to imatinib
Reply:
We acknowledged the reviewers’ comment and VEGFRI including sunitinib, regorafenib and pazopanib may decrease in their tumor size, but infrequently, as described in the review of Maki RG, Blay JY, Demetri GD, Fletcher JA, Joensuu H, Martín-Broto J, Nishida T, Reichardt P, Schöffski P, Trent JC. Key Issues in the Clinical Management of Gastrointestinal Stromal Tumors: An Expert Discussion. Oncologist 2015;20(7):823-30. Thus, we change as follows:
The SDH-GIST is resistant to all available tyrosine kinase inhibitors (TKIs) in most cases and may partly show transient stabilization or decrease in size under VEGFR inhibitor treatment because its progression is thought to be driven by the expression of insulin growth factor-1 receptor (IGF1R) and vascular endothelial growth factor receptor (VEGFR) induced by hypoxia-inducible factor-1α (HIF-1α).4,40
-lines 213-214: can you roughly quantify the risk of bleeding?
Reply:
Thank you for your comment. Ref. 45 (new Ref. 46) describes that one study showed that significant bleeding occurred in 35.7% of patients after jumbo biopsy, and 34.9% of patients needed subsequent endoscopic hemostasis. This value is from only one retrospective study and we do not describe this value in our review because it may mislead.
-line 329: local expertise also dictates the application of laparoscopy
Reply:
Thank you for your good indication. The details of laparoscopic application for large GISTs and GIST locating at challenging position, such as the EGJ, may be interesting, for example, for training surgeons, however, the application of laparoscopy is context-dependent, I mean, application may depend not only on tumors (size, form, location in the wall and of the organ) but also on persons and team. Thus, right now, there is no universal application of laparoscopy for GIST.
-line 390 and 488: “medicines” should be replaced with other word such as “treatments”
Reply:
We have changed “medicines”, one is to drugs and the other treatments.
-Figure 1: the word “Avapritib” is misspelled
Reply:
We are very sorry for miss spelling. Now, it is Avapritinib.

Reviewer 3 Report
The presented literature review is relevant. It presents information on the prevalence of histological tumors, the characteristics of their treatment and the prospects for further study.
Author Response
Reply from the authors to comments from the reviewers
Reviewer 3
The presented literature review is relevant. It presents information on the prevalence of histological tumors, the characteristics of their treatment and the prospects for further study.
Reply:
Thank you for the reviewer’s opinion.
